# Extracellular and intracellular intermittent magnetic-fluid hyperthermia treatment of SK-Hep1 hepatocellular carcinoma cells based on magnetic nanoparticles coated with polystyrene sulfonic acid

Bo-Wei Chen[1], Guo-Wei Chiu[2], Yun-Chi He[2], Chih-Yu Huang[2], Hao-Ting Huang[2], Shian-Ying Sung[3,4], Chia-Ling Hsieh[3,4], Wei-Chieh Chang[5], Ming-Shinn Hsu[6], Zung-Hang Wei[1,2], Da-Jeng Yao[1,2] *

1 Institute of NanoEngineering and MicroSystems, National Tsing Hua University, Hsinchu City, Taiwan, 2 Department of Power Mechanical Engineering, National Tsing Hua University, Hsinchu City, Taiwan, 3 Ph. D. Program for Translational Medicine, College of Medical Science and Technology, Taipei Medical University, Taipei, Taiwan, 4 TMU Research Center of Cancer Translational Medicine, Taipei Medical University, Taipei, Taiwan, 5 Department of Neurosurgery, Chang Bing Show Chwan Memorial Hospital, Changhua, Taiwan, 6 Department of Obstetrics and Gynecology, Ching-Kuo Campus of Min-Sheng Hospital, Taoyuan, Taiwan

* djyao@mx.nthu.edu.tw

**Data Availability Statement:** All relevant data are within the manuscript and its Supporting Information files.

## Abstract

The use of magnetic nanoparticles (MNPs) magnetized on applying an alternating magnetic field (AMF) to stimulate the thermal characteristics and to induce tumor apoptosis is a currently active area of research in cancer treatment. In previous work, we developed biocompatible and superparamagnetic polystyrene-sulfonic-acid-coated magnetic nanoparticles (PSS-MNPs) as applications for magnetically labeled cell trapping, but without assessment of treatment effects on tumor diseases. In the present work, we examined PSS-MNP-induced magnetic fluid hyperthermia (MFH) on SK-Hep1 hepatocellular carcinoma (HCC) cells for lethal thermal effects with a self-made AMF system; an adjustable AMF frequency generated a variable intensity of magnetic field and induced MNP relaxation. The extracellular and intracellular MFH treatments on a SK-Hep1 cell line were implemented *in vitro*; the result indicates that the lethal effects were efficient and caused a significantly decreased cell viability of SK-Hep1 cells. As the PSS-MNP concentration decreased, especially in intracellular MFH treatments, the MFH effects on cells, however, largely decreased through heat spreading to the culture medium. On controlling and decreasing the volume of culture medium, the problem of heat spreading was solved. It can be consequently expected that PSS-MNPs would be a prospective agent for intracellular cancer magnetotherapy.

**Funding:** The Ministry of Science and Technology (MOST), Taiwan, provided support for this study in the form of grants (MOST 105-2628-M-007-002-MY3 and MOST 104-2221-E-007-015-MY4) awarded to the following authors: BWC; GWC; YCH; CYH; HTH; ZHW; and DJY.

**Competing interests:** The authors have declared that no competing interests exist.

## Introduction

Hepatocellular carcinoma (HCC) is a common malignant disease and cause of cancer mortality [1]. Clinical surgical resection, chemotherapy, radiotherapy and thermotherapy have been applied for many years [2], but the side effects of tumor recurrence and possible mitigating metastases after treatment remain main issues of the problems to be solved [3]. In recent years, magnetic nanoparticles (MNPs) have attracted significant attention as a formulation of cancer treatment with diminished side effects, because of the remote controllability and possible penetration through the intact vessel fenestration to reach the tumor tissues and organs, such as nanocarriers for drugs or biochemical molecules in local hyperthermia [4] and magnetic targeting [5] and contrast-imaging agents in magnetic-resonance imaging (MRI) [6–8].

Magnetic-fluid hyperthermia (MFH) treatment is a thermotherapy of a kind utilizing MNPs that accumulate in the tumor tissues and generate a friction contributed from the alternating-magnetic-field(AMF)-induced magnetic-moment spin flipping (Néel relaxation) and nanoparticle rotation (Brownian relaxation) to provide local heat to the surrounding tissues or organs [9]. This heat can raise the local tissue temperature above 43˚C and induce the cancer cells to undergo apoptosis or necrosis without obvious damage to normal cells. It has also been proved that the thermal effects can inhibit the recovery of tumor cells from DNA damage [2]. Conversely, the normal cells can remain intact during thermotherapy; the side effects in hyperthermia treatments are thus less than those of chemotherapy or radiotherapy [10].

In addition to the MFH-induced lethal thermal effects, the mechanical destruction contributed from the MNPs Brownian relaxation induced by AMF induction is considered to provide one available method to improve the therapeutic efficiency that exerts a significant influence on cell proliferation and results in a cell-apoptosis mechanism. MNP-based treatments are divisible into sessions of three types according to the AMF operating frequency: high-frequency (300–1100 kHz), low-frequency (<100 kHz) and ultra-low-frequency AMF-induced hyperthermia [11, 12]. The high-frequency AMF-induced hyperthermia was developed a few decades ago; its therapeutic effects are contributed mainly from Néel relaxation [9], which could be enhanced and improved on conjugating nanodrugs onto the MNPs [13]. As the AMF operating frequency increases, the AMF-induced MNPs can effectively provide heat to kill the cells with more efficient lethal effects. An inappropriate control of the MNPs exothermal distribution and dissipating excess energy are, however, negative issues for human health in the related therapies; the risk of improper operation causing side effects should thus be avoided as much as practicable. The specific loss power (SLP) and intrinsic loss power (ILP) of the MNPs should be studied carefully; alternatives should be developed [14].

Under the restrictions of low-frequency AMF operation, the mechanical destruction through nanoparticle oscillation has attracted much attention: the use of MNPs to destroy cancer cells using external mechanical forces transmitted by low-frequency AMF-induced vibrations is considered a promising method to implement [11, 12]. Shi et al. noted that anisotropic rod-shaped (length 200±50 nm, diameters 50–120 nm) and spherical (diameter 200±50 nm) iron-oxide ($Fe_3O_4$) nanoparticles have been developed and used for effective cell killing under 35-kHz AMF [11]; in comparison with spherical MNPs, the low-frequency AMF-induced rod-shaped MNPs provides an effective torque on the human cervical cancer-cell line (HeLa), indicating that low- frequency AMF induction might become a key technique for cancer therapy with major therapeutic effects and minor adverse effects. Because of the small mass fraction of MNPs in the cells, the temperature rise is too small, however, to provide effective hyperthermia; multifunctional applications with both mechanical and thermal lethal effects are thus difficult to achieve. Similar results were observed by Leulmi et al.: ferromagnetic nanoplates (diameter 1 μm, thickness 60 nm) were prepared through a top-down approach, cultured to a

human SKRC-59 renal cancer-cell line and magnetized under ultra-low AMF (frequencies $\sim 20$ Hz and weak magnetic fields $\sim 30$ mT) in turn [15]. The anisotropic nanoplates provide no significant thermal effect because of the low operating frequency, indicating that the lethal effects are contributed mainly from the mechanical force and torque. The low-AMF control to integrate a mechanical-force-based and magnetic hyperthermia treatment is still a challenging topic in multifunctional therapies and applications.

In this work, to integrate the mechanical and magnetic hyperthermia effects of MNPs for multifunctional therapy applications, we constructed a simple AMF instrument to adjust the output magnetic field at a stable alternating frequency. The instrument is comprised of an iron core wrapped in a copper coil, a resonant RLC- circuit power supply and a circulating water bath. According to our tests to evaluate this instrument, it can generate a magnetic field 25–160 G with AMF frequency 40–170 kHz. Moreover, we have developed superparamagnetic magnetic nanoparticles coated with polystyrene sulfonic acid (PSS-MNPs) as a kind of hydrophilic and magnetically controllable biomaterial for applications of manipulating magnetic cells in concentric magnetic structures [16]. The satisfactory biocompatibility of these PSS-MNPs has been proved also through MNP co-incubation with human SK-Hep1 HCC and mouse NIH-3T3 fibroblast cell lines [17]. There is no report about the treatment efficiency and effects of PSS-MNP AMF-based therapies in human HCC cancer treatments. The low-frequency AMF induction and strong flux of the magnetic field can be provided to assess PSS-MNPs in multifunctional biomedical treatments. The primary biomedical analyses of PSS-MNPs were assessed first through a MTT assay and Prussian-blue staining for cell viability and nanoparticle uptake efficiency, respectively, in SK-Hep1 HCC cell lines. The hyperthermia tests were divided into two sessions, extracellular and intracellular MFH, to evaluate the PSS-MNP treatment effects and their SAR. The nuclei of MFH-treated cells were observed, under DAPI/PI staining observation, to undergo the physical characterization for an integrated influence of the mechanical force and thermal lethal effects on the HCC cells. The characterization of PSS-MNPs and optimal AMF intensity and frequency in these HCC thermotherapies should be understood for the applicability of PSS-MNPs.

## Materials and methods

### Chemicals, reagents and cell lines

For MNP preparation, iron(III) chloride (97%, $FeCl_3.6H_2O$, Showa, Japan), iron(II) sulfate (98%, $FeSO_4.7H_2O$, Showa, Japan), polystyrene sulfonic acid (PSS, 30% w/v, molecular mass 75000, Alfa Aesar, UK) and aqueous ammonia ($NH_4OH$, 25% v/v, Sigma Aldrich, USA) were obtained from the indicated sources.

The human-liver adenocarcinoma SK-Hep1 (ATCC® HTB-52TM) cell line was used in this work. For cell culture, Dulbecco´s modified Eagle medium (high glucose, with L-glutamine 4 mM), fetal bovine serum (FBS, analytical reagent grade), penicillin/streptomysin solution (10000 units $mL^{-1}$ penicillin/streptomycin in NaCl, 0.85%), 10×trypsin, 10×phosphate-buffered saline (PBS) and 4',6-diamidino-2-phenylindole (DAPI) (Thermo Scientific, Fisher Scientific, USA) and glutaraldehyde (Polysciences, USA), 3-(4,5-dimethylthiazol-2-yl)-2,5-diphenyltetrazolium bromide (MTT, Alfa Aesar, UK), potassium hexacyanoferrate ($K_4[Fe(CN)_6$, 99%, Showa Chemical, Japan), propidium iodide (PI) (stock 10 μg $mL^{-1}$, Invitrogen, USA) and dimethyl sulfoxide (DMSO, $\geq$99.9%, analytical reagent grade, J. T. Baker, USA) were obtained from the indicated companies. Commercial iron-oxide product EMG705, (Ferrotec Corp.) served for concentration calibration for PSS-MNPs on Prussian-blue staining examination.

## Preparation of hydrophilic magnetic nanoparticles coated with PSS (PSS-MNPs)

The preparation of MNPs is reported elsewhere [17]. In brief, PSS (5 mL, 30% w/v) was dissolved in distilled water (50 mL) under constant stirring for 2 h under a $N_2$ atmosphere to obtain PSS micelles. Thereafter, aqueous solutions of $FeCl_3$ (10 mL, 1 mM) and $FeSO_4$ (10 mM, 0.5 mM) were rapidly injected into the above mixture. Following heating to 65°C, aqueous $NH_3$ (7 mL) was added; the reaction mixture turned black, which indicated the formation of MNPs. The reaction was allowed to proceed for 5 h under vigorous stirring (1100 rpm) until PSS-stabilized magnetite nanoparticles were obtained. The resulting mixture was washed twice with distilled water and was preserved for further applications.

## Assembly of hyperthermia system

For hyperthermia tests, we assembled a magnetothermal setup consisting of a resonant RLC-circuit machine connected to a copper coil capped with enamel insulated wire (diameter 0.45 mm) to perform the hyperthermia experiments (Fig 1A; detailed photograph of instruments in S1 Fig). The copper-coil-wrapped iron core ($N = 100$) was placed in an acrylic cylinder and separated with water that circulated from the water bath to maintain the temperature constant at 37°C. This device can serve as an alternating-current (AC) system of variable frequency and generate an AMF of variable intensity. An oscilloscope (InfiniiVision DSO5012A, Agilent Technologies, USA) was used to adjust and to measure the output power at a varied frequency from the RLC circuit (at 43.7, 70 and 143.5 kHz). The intensity of the generated magnetic field was detected with a facile magnetometer and compared with the results of output power in the oscilloscope.

## Cell culture and magnetic labeling of PSS-MNPs on the SK-Hep1 cell line

Before the cell culture, the exosome-depleted FBS was obtained through its centrifugation at rate 2000 rpm for 20 min, to deplete the bovine exosomes. The SK-Hep1 cell line was cultured and seeded in culture dishes (30 mm) with DMEM (1 mL), which was supplemented with exosome-depleted FBS (10%), penicillin (100 unit $mL^{-1}$) and streptomycin (100 units $mL^{-1}$). The dishes were placed in the acrylic cylinder of a hyperthermia system in an incubator (37°C, atmosphere 5% $CO_2$) for 24 h, to attain 90% confluence. After that 24-h cell culture, the cell solution was washed with 1×fresh PBS 2–3 times and treated with 1×trypsin (1 mL) at 37°C under an atmosphere with 5% $CO_2$ for 3 min, followed by removal of trypsin by centrifugation and addition of fresh FBS medium. The floating-cell solution was divided into a 96-well plate (0.1 mL/well, density $5 \times 10^5$ cells $mL^{-1}$) for further characterization and applications.

For the co-culture with PSS-MNPs, the cells in the 96-well plate were treated with PSS-MNPs at concentrations 0.01, 0.05, 0.1, 0.5, 1, 5 and 10 mg $mL^{-1}$ for 24 h. The MNP-treated cells were washed three times with fresh PBS to remove excess MNPs. The cell viability was thus detected and characterized with a MTT assay. The cell viability is expressed as the percentage of the absorption of cells treated with MNPs relative to control cells.

## MTT assay for examination of cell viability

MTT in PBS was prepared at stock concentration 0.5 mg $mL^{-1}$. This stock solution was added to the cell solution in volume ratio 1:1 (v/v); the cells were incubated in the incubator for 4 h (37°C, 5% $CO_2$ atmosphere). After incubation, the supernatant in each sample was removed carefully, and immersed in DMSO. After mild shaking (10 min), the purple formazan products were uniformly dissolved in DMSO. An enzyme immunoassay (ELISA) microplate reader

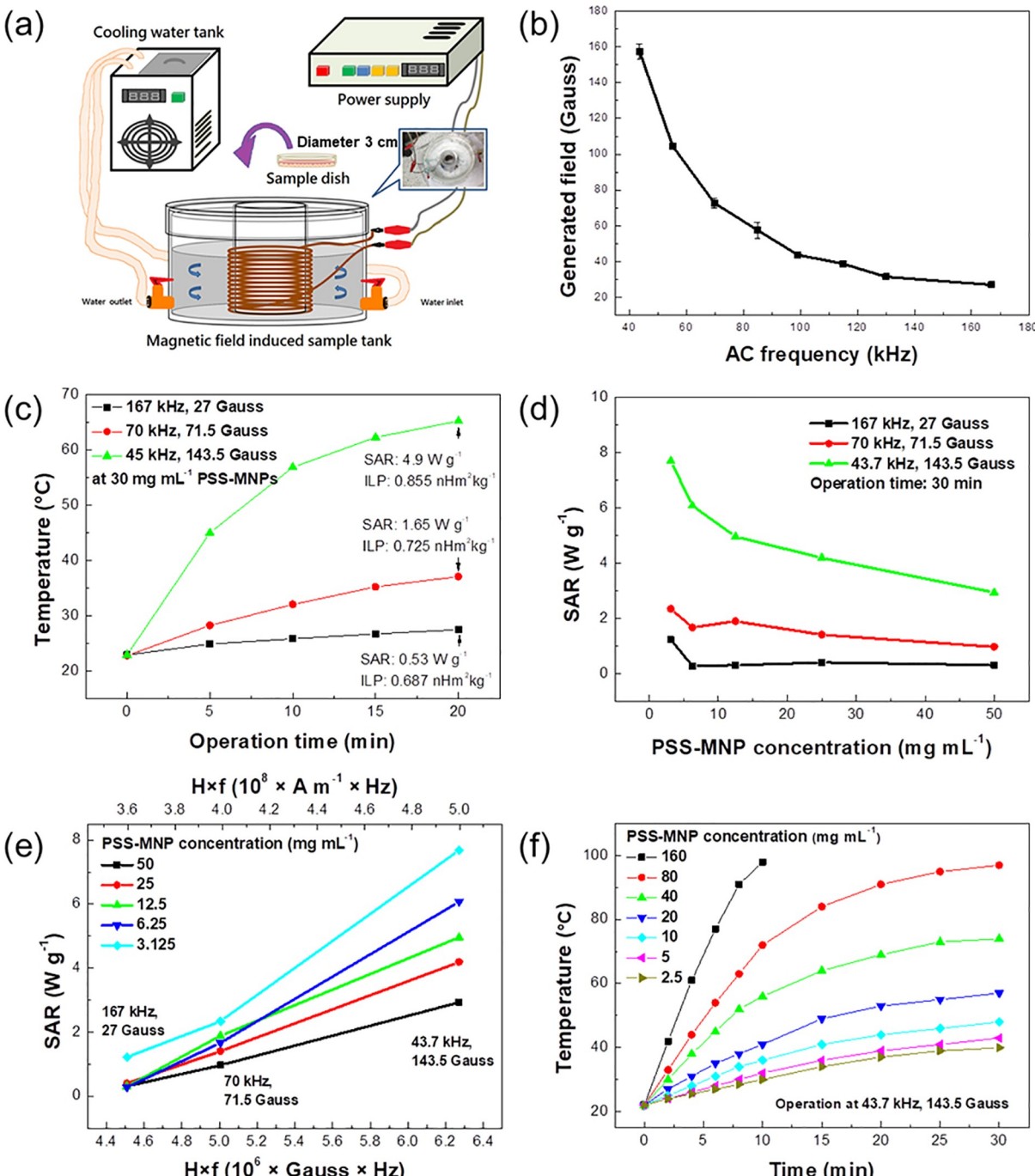

**Fig 1. Performance of our AMF system and results of examination of the heating efficiency of PSS-MNPs.** (a) Schematic device of high-frequency AMF-induced heating system. The system contains of three main parts—resonant RLC circuit machine, tank for cooling with circulated water and magnetic-field-induced sample tank. (b) Magnetic-field relative-frequency curve of our AMF device. (c) Time-dependent temperature (TDT) curve of PSS-MNPs (30 mg mL$^{-1}$) under varied intensity of applied AMF (27, 71.5 and 143.5 G) for 20 min from initial temperature 23 $^o$C. SAR and ILP values are labeled in the figure. (d) SAR *vs* PSS-MNP concentration (3.125, 6.25, 12.5, 25 and 50 mg mL$^{-1}$) for varied intensity of applied AMF (27, 71.5 and 143.5 G) for 30 min. (e) SAR values *vs* $H \times f$ (product of generated AMF and operating frequency, generated AMF intensity s$^{-1}$) with PSS-MNP concentrations 3.125, 6.25, 12.5, 25 and 50 mg mL$^{-1}$ under hyperthermia operation for 30 min. (f) TDT curve of culture medium with added PSS-MNPs of concentration 2.5, 5, 10, 20, 40, 80 and 160 mg mL$^{-1}$ in the case of a fixed AMF (43.7 kHz, 143.5 G) for 30 min.

(Thermo Scientific, Fisher Scientific, USA) was used to measure the absorbance of samples at wavelength 570 nm.

## Examination of endocytosis efficiency of PSS-MNPs on the SK-Hep1 cell line via Prussian-blue staining

A commercial magnetic iron-oxide nanoparticle solution (EMG 705) at varied concentration was used to calibrate a standard curve for the MNP concentration. This EMG 705 MNP solution was dried with baking at 110˚C to remove the aqueous solvent and moisture, followed by addition of HCl solution (0.5 mL, 5 M) at 60˚C for 4 h to convert iron oxide to $Fe^{3+}$ ions to form a yellow solution. Before adding an equal amount of $K_4[Fe(CN)_6]$ solution (5% w/w) to the iron ion solution, the solution turned from yellow to blue because of the formation of blue stains. After 40 min to complete the reaction between iron and $K_4[Fe(CN)_6]$, the absorbance at 620 nm was read with an ELISA microplate reader; the absorbance at each concentration measured on the PSS-MNP concentration served to produce a standard curve.

Similarly, the cells were co-cultured with PSS-MNPs (0, 0.01, 0.05, 0.1, 0.5 and 1 mg mL$^{-1}$) for 24 h and centrifuged at 1000 rpm for 5 min. The supernatant was removed; about 5 x 10$^5$ cells were dispersed in a phosphate buffer solution (1 mL) and dried in an oven at 110˚C, followed by addition of HCl (0.5 mL, 5 M) at 60˚C for 4 h to convert iron oxide to $Fe^{3+}$ ions. An equal amount of $K_4[Fe(CN)_6]$ solution (5% w/w) was added and mixed, and reacted at room temperature for 40 min. After the color became stabilized, the absorbance at 620 nm was read with the ELISA microplate reader. The absorbance values were converted into an uptake concentration of PSS-MNPs according to the standard curve from the EMG 705 and the thermogravimetric analysis (TGA) of PSS-MNPs in a previous study [17]. The uptake concentration in cells measured at each concentration was plotted against the PSS-MNP concentration. The uptake concentration is divided by the total number of cells to obtain the PSS-MNP uptake concentration of a single cell.

## Hyperthermia experiments performed on cells

**I. Heating efficiency of PSS-MNPs.** After the preparation of PSS-MNPs, their solution at various concentrations and for varied duration of operation for primary heating efficiency was evaluated on applying the AMF system through a time-dependent analysis. The temperature was detected with a thermocouple. The experimental setup is shown in S1 Fig.

**II. Cell pretreatment of hyperthermia experiments.** The experimental steps were similar to the descriptions in the previous part. Cells were seeded at concentration 5×10$^5$ cells/dish (medium volume 1 mL) in culture dishes (30 mm) at 37˚C under 5% $CO_2$. After incubation for 24 h, cells were treated with PSS-MNPs at varied concentration for the hyperthermia experiments. Before the *in vitro* hyperthermia treatments, the AMF system was preheated and maintained in an incubator (37˚C, 5% $CO_2$).

Herein, **(1)** in the extracellular magnetotherapy treatments, after cells were treated with PSS-MNPs (5 or 10 mg mL$^{-1}$) and incubated for 24 h, the cells were immediately moved onto the AMF setup structure and the AMF was applied. **(2)** For intracellular hyperthermia examinations, cells were treated with PSS-MNPs (0.05 or 0.5 mg mL$^{-1}$), and incubated (24 h) for cell endocytosis. The dose concentrations were chosen on the basis of results from Prussian-blue staining such that the PSS-MNP uptake attained a maximum internalization value when the cells (SK-Hep1 cell line) were treated with PSS-MNPs (0.05 or 0.5 mg mL$^{-1}$). The incubated cells were washed twice with PBS until the black-grey particles and medium solution were removed before the AMF application. After AMF treatments, the cells were further characterized with a MTT assay to examine the cell viabilities.

### DAPI/PI staining

Cells were fixed with paraformaldehyde (PFA, 3.7%) for 10 min and then washed in PBS. The cell nuclei were stained with DAPI (1:1000 dilute) and PI (1 μL, 10 μg mL$^{-1}$), incubated in the dark for 30 min and sequentially analyzed with a fluorescence microscope (Olympus CKX41). Cells were imaged with an inverted fluorescence microscope equipped with blue, green and UV filters. The wavelength of the divergent light was about 400 nm; the wavelength of maximum absorption was 358 nm; the wavelength of maximum emission was 461 nm. All experimentally reported data were produced in triplicate.

### Data analysis

Data analysis was performed (Origin 85 Pro, OriginLab Corporation, Northampton, MA, USA). Three independent tests were performed to obtain the final results. The values acquired from experiments in series are represented as mean ± standard deviation (SD). At least Student's *t* test was used to compare the cell viability between PSS-MNP untreated, PSS-MNP treated and hyperthermia-treated cell groups on the SK-Hep1 HCC cell line; the significance level was set at $p < 0.05$, $p < 0.01$ and $p < 0.001$.

## Results

### Properties of PSS-MNPs and AMF system building

In previous work, the prepared hydrophilic PSS-MNPs showed the features of satisfactory saturation magnetization (60 emu g$^{-1}$ at 300 K), nanoscale size (average 11 nm, measured with a transmission electron microscope TEM), appropriate monodispersity (hydrodynamic size in PBS 75–250 nm, average 130 nm, polydispersity index 0.0039), and high uptake efficiency and low cytotoxicity in human SK-Hep1 and mouse NIH-3T3 cell lines [17]. We demonstrated that the rate of cellular uptake of PSS-MNPs in SK-Hep1 was significantly greater than in NIH-3T3 cells, indicating a possible tumor specificity to the HCC cells in the absence of a conjugated antibody on the MNP surface [17]. The PSS-MNP property summary is shown in S2 Fig (TEM image) and S1 Table.

In the present study, for a MFH evaluation of PSS-MNPs in HCC cells *in vitro*, we built an AMF system shown in Fig 1A and S1 Fig. The system consists of a hollow- cylinder-shaped acrylic structure, a resonant RLC-circuit machine (AC power supply), a circulation tank for water cooling (constant temperature) and an iron cylinder core wrapped with copper coil ($N = 100$) (a self-made electromagnet). The resonant RLC-circuit machine with variable operating frequency provided a variable generated magnetic field from the AMF system, showing that the generated AMF intensity decreased as the operating frequency increased (Fig 1B).

To evaluate the heating efficiency and SAR contributed from PSS-MNP relaxation under varied AMF control parameters, PSS-MNPs (1 mL, 30 mg mL$^{-1}$) were applied with operating efficiency 27×167, 71.5×70 and 143.5×43.5 kG Hz for 20 min at temperature 23 °C (Fig 1C). SAR and ILP values as the heat dissipation by PSS-MNPs expressed as W g$^{-1}$ and H m$^2$ kg$^{-1}$ were calculated; the period 20 min was considered for this SAR and ILP calculation:

$$\text{SAR} = (C_{\text{solvent}})/((X_{\text{magnetic element}}) \times (\mathrm{d}T/\mathrm{d}t)),$$

in which $C_{\text{solvent}}$ is the heat capacity of water, 4.18 J g$^{-1}$ K$^{-1}$, $X_{\text{magnetic element}}$ is the mass fraction of MNPs in the solution (g L$^{-1}$) and $\mathrm{d}T/\mathrm{d}t$ is the slope of the curve of temperature change (K or °C) *versus* time (s) [18].

$$\text{ILP} = \text{SAR}/(f \times H^2),$$

in which $f$ is the AMF operation frequency and $H$ is the magnetic field generated from the AMF system [19].

At operating frequency 43.5 kHz, the temperature, SAR and ILP reached 65 °C, 4.9 W g$^{-1}$ and 0.885 nH m$^2$ kg$^{-1}$, respectively; this heating efficiency was better than those at operating frequency 167 kHz (36 °C, 1.65 W g$^{-1}$, 0.687 nH m$^2$ kg$^{-1}$) and 70 kHz (26 °C, 0.53 W g$^{-1}$, 0.725 nH m$^2$ kg$^{-1}$) (Fig 1C). Although the SAR values decreased gradually as the PSS-MNP concentration increased (Fig 1D and 1E), PSS-MNPs at 5 mg mL$^{-1}$ sufficed to attain temperature 42 °C from 23 °C in 30 min under AMF operation at 143.5 G, 43.5 kHz (Fig 1F).

Superior SAR values were hence obtained when the PSS-MNP concentration was about 3–10 mg mL$^{-1}$; the MNP concentration affected the performance of SAR significantly (Fig 1D). In the larger $H \times f$ state (product of generated AMF and operating frequency: generated AMF intensity s$^{-1}$), the SAR value increased (Fig 1E); to increase the heating efficiency of iron oxide is hence a feasible method to increase further the magnetic field output ($H \times f$ state) of the AMF system.

## Biocompatibility analysis of PSS-MNPs in human hepatocellular carcinoma (HCC) cells

To assess the availability of PSS-MNPs for the HCC cancer therapy, we selected the SK-Hep1 cell line to examine the primary *in vitro* biocompatibility. The SK-Hep1 cells were co-incubated with PSS-MNPs (0, 0.05, 0.1, 0.5, 1, 5 and 10 mg mL$^{-1}$) separately for 24 h, followed by implementing a MTT assay and Prussian-blue staining to test the biocompatibility and internalization efficiency, respectively. As the MTT assay showed, the rate of cell survival was still more than 90%, which shows that the PSS-MNPs showed no significant cytotoxic effect on the SK-Hep1 cell line (Fig 2A). The efficiency of cell internalization of MNPs indicated maximum endocytosis values 70 μg mL$^{-1}$ for SK-Hep1 cells that had been treated with PSS-MNPs (0.5 mg mL$^{-1}$) (Fig 2B).

## Extracellular hyperthermia in human HCC cells for PSS-MNP evaluation

After the SK-Hep1 cells were co-incubated with PSS-MNPs (5 or 10 mg mL$^{-1}$) for 24 h, the cells were treated with MFH for 0.5, 1, 2 and 3 h. The cell viability values of MNP-treated cells significantly decreased to 61.13% and 33.71% (PSS-MNP concentrations 5 and 10 mg mL$^{-1}$, respectively) for SK-Hep1 cells, indicating that PSS-MNP-based MFH performed an effective killing effect on the HCC cells relative to the control groups without MFH treatment (Fig 2C). Notably, the MFH effect after 3 cycles (1 cycle: MFH 1 h, off 30 min) effectively decreased the cell viability in SK-Hep1 cells (30.68%; Fig 2D); as the MFH operating duration and PSS-MNP concentration increased, the inhibition of tumor cells became more efficient.

Fig 3 shows optical and fluorescent images of DAPI/PI-stained HCC cells on performing the clear DAPI and PI staining of the nuclei and dead cells, respectively. It is significant that the cells untreated and treated with MFH exhibited distinct results on cell proliferation: MFH-untreated SK-Hep1 cells in Fig 3 show similar results with Fig 2A because of a lack of PI stained cells under fluorescence observation, indicating that the PSS-MNPs exhibited a low cytotoxicity. In contrast, most cells that underwent MFH for 1 h were stained with PI, implying detrimental effects and damage to the HCC liver-cancer cells. In addition, the appearance of the cell-MNP-mixture solution in an Eppendorf tube showed an obvious change of their dispersity (Fig 3C); the MFH-treated cells tended to shrink, undergoing a decreased cell ductility and aggregating with each other (ii and iii in Fig 4A and 4B). Under enlarged fluorescent images (Fig 4B), few DAPI-stained nuclei in MFH-treated cells broke into fragments (Fig 4B (v, vi)), resulting in intracellular matrix damage and possible apoptotic reaction due to heat

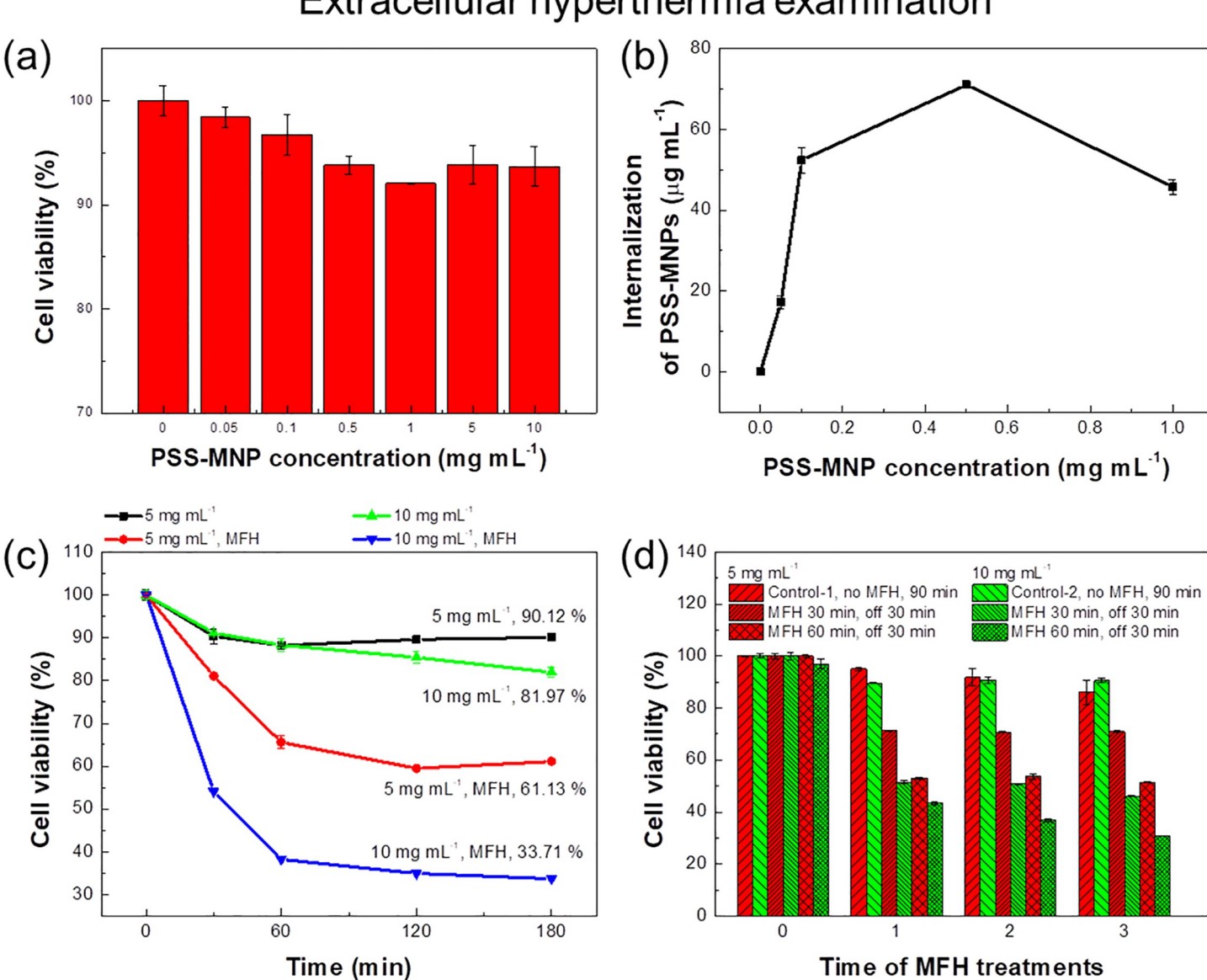

**Fig 2. *In vitro* examination of a SK-Hep1 cell line to evaluate PSS-MNPs.** (a) SK-Hep1 cells (5×10⁵ cell mL⁻¹) were incubated with PSS-MNPs at 0, 0.05, 0.1, 0.5, 1, 5 and 10 mg mL⁻¹, respectively. After incubation for 24 h, cell viability was tested with MTT analysis. (b) Examination of internalization efficiency of PSS-MNPs in a SK-Hep1 cell line through application of Prussian-blue staining. (c, d) Extracellular hyperthermia effects on a SK-Hep1 cell line induced by PSS-MNPs (concentrations 5 and 10 mg mL⁻¹) after high-frequency AMF treatments (43.7 kHz, 143.5 G) for operating durations 0.5, 1, 2 and 3 h. Control groups in (c) and (d) mean the cells were untreated with PSS-MNPs. (e, f) Extracellular hyperthermia effects on a SK-Hep1 cell line induced by PSS-MNPs (concentrations 5 and 10 mg mL⁻¹) after high-frequency AMF treatment (43.7 kHz, 143.5 G) for 0–3 cycles. The definition of cycle in hyperthermia-treated cell lines was set as the hyperthermia operation period 30 or 60 min with 30 min machine off; the definition of a cycle in a control group means that a MTT assay was implemented after the cells were co-cultured with PSS-MNPs for 1.5 h. Control groups in (e) and (f) mean the cells were treated with PSS-MNPs and cultured for varied duration of incubation, which was the duration of incubation with the hyperthermia examination and the product of duration of treatment operated.

influence and the mechanical force by magnetized PSS-MNPs. The cell size of the SK-Hep1 cells moreover decreased posterior to the MFH treatments (Fig 4C), accompanied with deformation of the cell nuclei. The negative effects on cell growth and proliferation were hence contributed from MFH treatments.

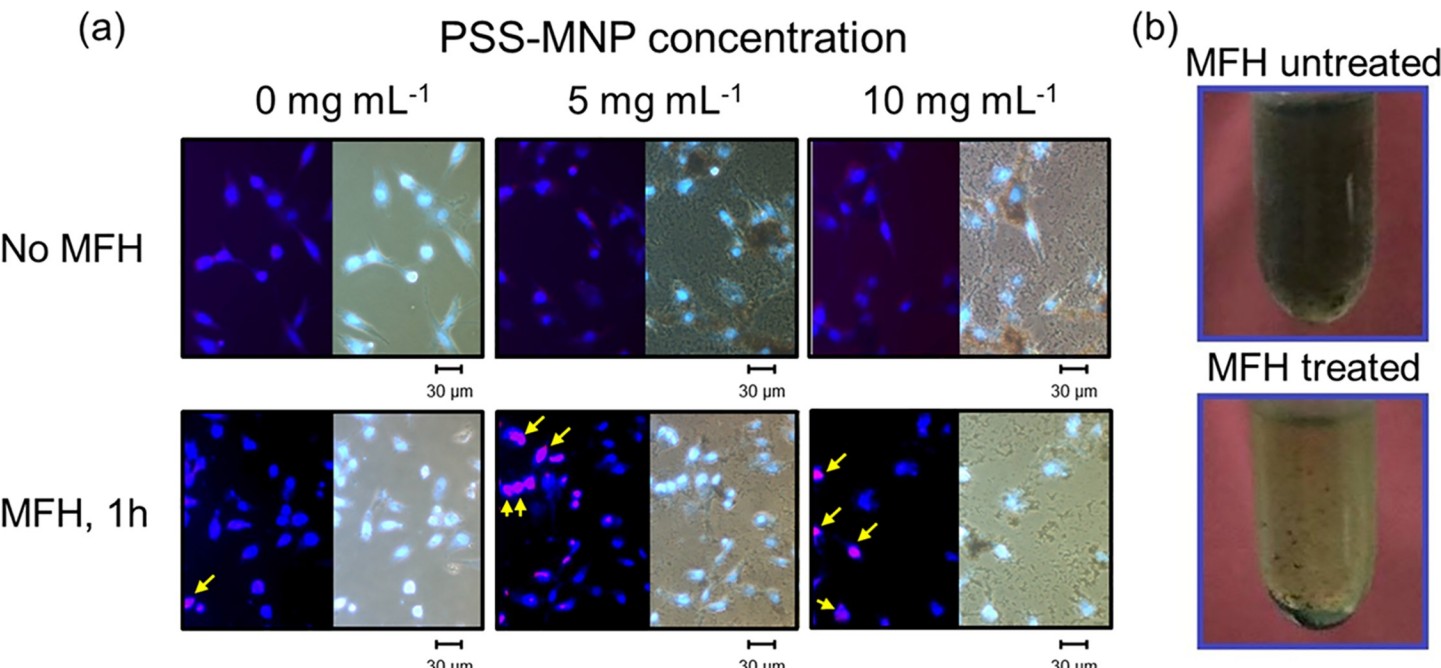

**Fig 3. Cell morphologies and appearance of DAPI/PI-stained cells.** (a) Observations of SK-Hep1 cell line after incubation for 24 h with PSS-MNPs under optical and fluorescence microscopes (merged) and (b) photographs of SK-Hep1 cells treated with PSS-MNPs before and after 1 h of extracellular hyperthermia treatment. The scale bar is 30 μm.

### PSS-MNP induced intracellular hyperthermia effects in human HCC cells

Dependent on the characterization of the efficiency of PSS-MNP endocytosis uptake and bio-distribution in HCC cells (Figs 2B and 3), the concentration of PSS-MNP saturation uptake treated in SK-Hep1 cells was 0.5 mg mL$^{-1}$. In intracellular MFH treatment the cells that were co-cultured with PSS-MNPs (0.5 mg mL$^{-1}$) were implemented to study the therapeutic effects of MNPs inside SK-Hep1 cells. Contrary to the previous statement, the intracellular MFH effects on SK-Hep1 cells had, however, no obvious MFH effects, resulting in cell-viability values about 85% (Fig 5A). Whether the cells were in an unsaturated or saturated state of the MNP concentrations (0, 0.5, 5 mg mL$^{-1}$), the thermal lethal effects were insignificant, perhaps because of the decreased amount of effective endocytosed MNPs in cells. As another reason, there might be a cooling effect contributed by the extracellular culture medium, causing an easing of thermal effects. In Fig 5B, the SK-Hep1 cells were treated with PSS-MNPs (0.5 mg mL$^{-1}$), followed by intracellular MFH treatments with varied solvent volume (0.2, 0.4, 0.6, 0.8, 1 mL medium), which shows that treatment with less solvent (culture medium) exerted more efficient thermal lethal effects.

### Intermittent PSS-MNP induced intracellular hyperthermia effects and the influences of secondary administration on human HCC cells

The MNP-treated SK-Hep1 cells underwent further AMF operations for a few cycles. The definition of 1 cycle in Fig 6 is that the cells were treated with AMF operation for 2 h with 0.5 h rest in a medium (volume 0.2 mL) at 37°C and 5% $CO_2$ atmosphere. As the operating cycles increased, the therapeutic effect on SK-Hep1 HCC cells improved. The cell viability of SK-Hep1 cells after three cycles of AMF induction exhibited 65% (Fig 6A), indicating that possible apoptosis and extracellular matrix damage occurred as the result of hyperthermia

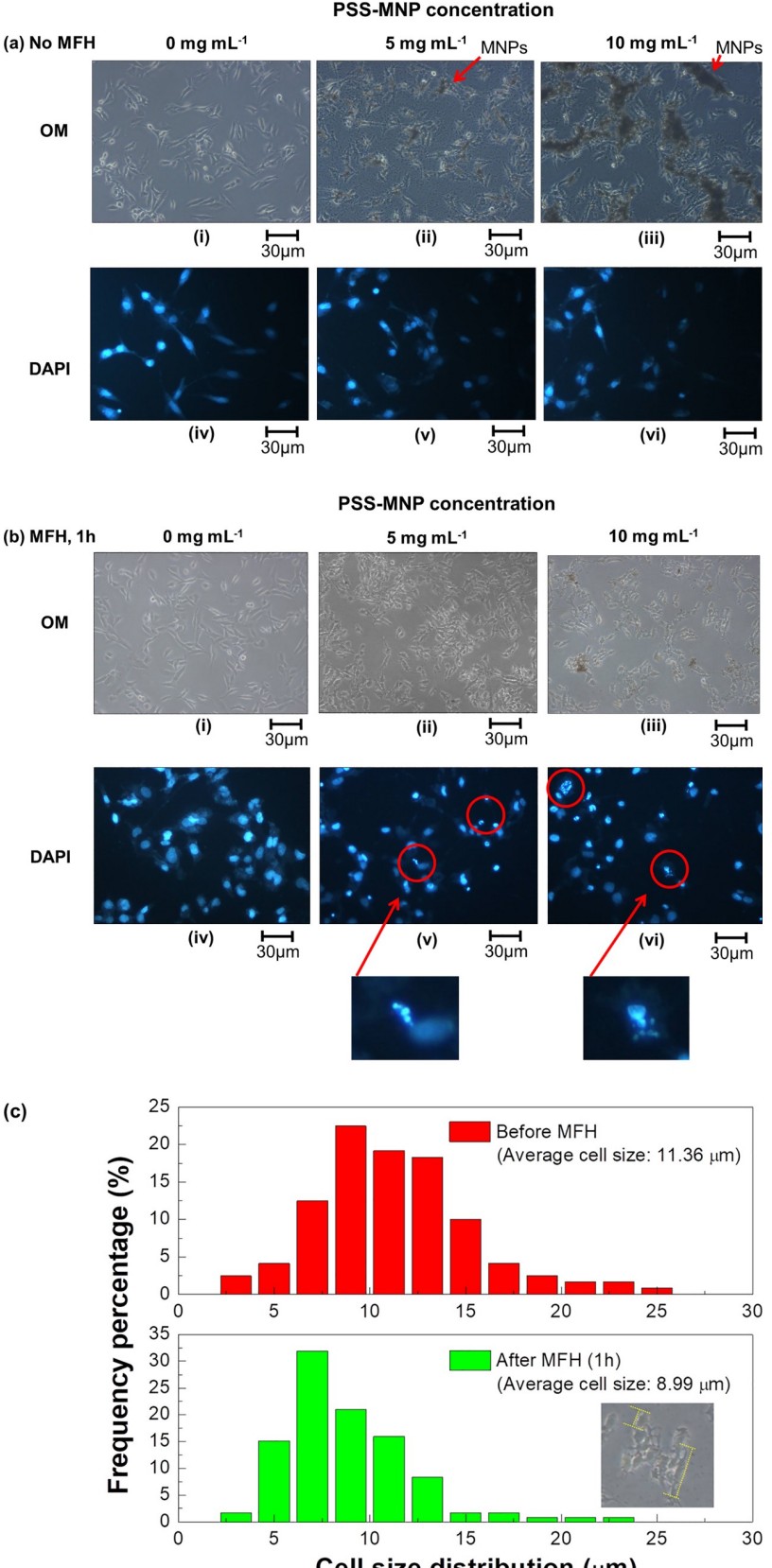

**Fig 4. Images and distribution of cell size of DAPI-stained SK-Hep1 cells treated or untreated with MFH.** DAPI-stained MNP-treated SK-Hep1 cells were (a) untreated or (b) treated with MFH for 1 h; size comparison of SK-Hep1 cells before and after MFH operation (1 h) after co-culture 24 h with PSS-MNPs (10 mg mL$^{-1}$). The SK-Hep1 cells were incubated with PSS-MNPs (i, iv) 0, (ii, v) 5 and (iii, vi) 10 mg mL$^{-1}$ for 24 h, followed by PBS washing three times to remove the waste and excess PSS-MNPs from the medium. The cell solutions were then (a) observed under optical and fluorescence microscopes or (b) treated with hyperthermia for 1 h before microscope observation. Scale bars in figures are 30 μm. (c) Differences in size distribution of SK-Hep1 cells co-cultured with PSS-MNPs (10 mg mL$^{-1}$) for 24 h before and after MFH (1 h). The inset figure presents an enlarged image of MFH-treated SK-Hep1 cells with a definition of measured cell size (yellow dashed line), which labels the maximum cell size of the calibrated individual single cells.

treatment after PSS-MNP administration. After AMF inductions, the cells were immediately treated with PSS-MNPs (0.5 mg mL$^{-1}$) as a secondary administration for 24 h, followed by a few cycles of AMF applications (Fig 6B). The SK-Hep1 cells that experienced such a secondary administration exhibited a worse cell viability, from 65% to 55%, after incubation for 24 h; the cells attained cell viability 43% after a further two cycles of MFH treatment.

## Discussion

SK-Hep1 cells derived from a patient with liver adenocarcinoma were treated as a human HCC cell-line model because of their invasive and metastasic nature [20]. Recent studies indicate that the SK-Hep1 cells express mesenchymal stem-cell (MSC) markers without a performance of liver function but an endothelial function of endocytosis and tubular formation [21, 22]. The proposal to categorise the SK-Hep1 cells as liver sinusoidal endothelial cells but not as the HCC cell line is thus increased. The tumorigenesis, aggression and metastasis from

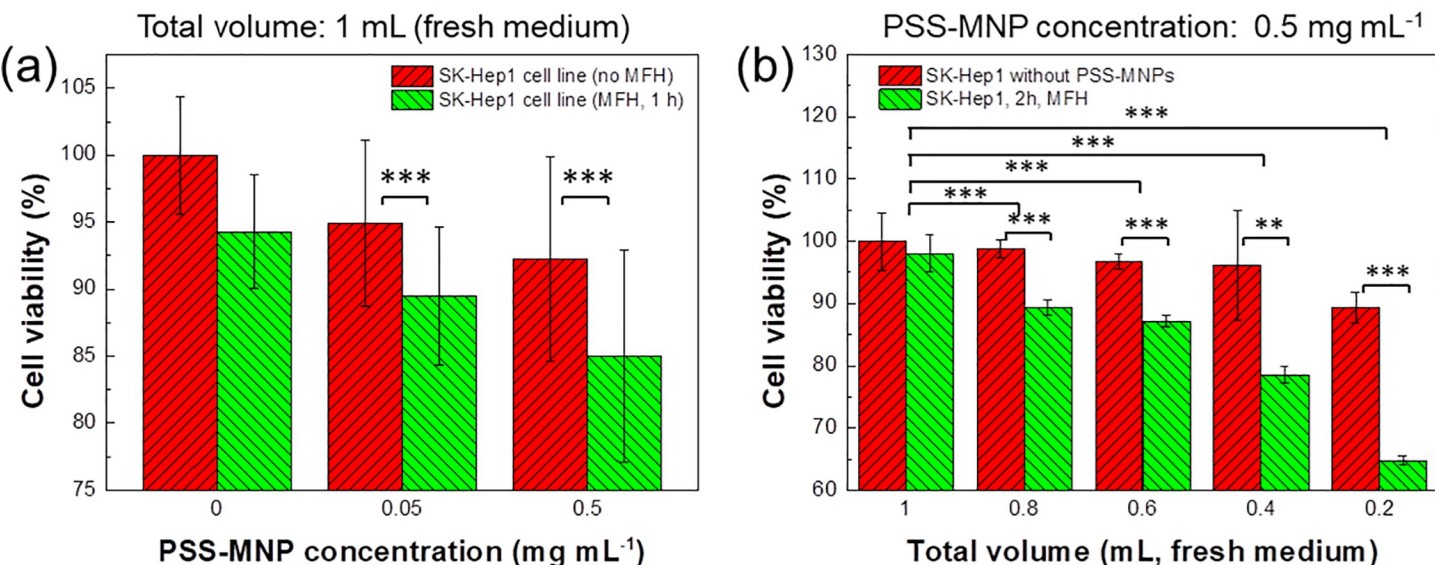

**Fig 5. Cell viability examination for intracellular MFH effects on SK-Hep1 cells with varied concentration and volume.** (a) Hyperthermia effect on HCC cells induced by intracellular MNPs (concentrations 0.05 and 0.5 mg mL$^{-1}$) after high-frequency AMF hyperthermia for 1 h. The control groups indicate that the samples were treated with an AC magnetic field without added MNPs. (b) Intracellular hyperthermia magnetotherapy effects in a culture medium of varied volume. The cells were treated with PSS-MNPs (0.5 mg mL$^{-1}$) for 24 h; untreated cell lines were incubated as control groups. Both MNP-treated and untreated cells were washed with PBS and filled with fresh culture medium (0.2, 0.4, 0.6, 0.8 and 1 mL) before AMF induction. Cells were treated for 2 h with AMF, at magnetic field 157 G and frequency 43.7 kHz. The degree of significance is given as $^{*}$ $p<0.05$; $^{**}$ $p<0.01$ and $^{***}$ $p<0.001$.

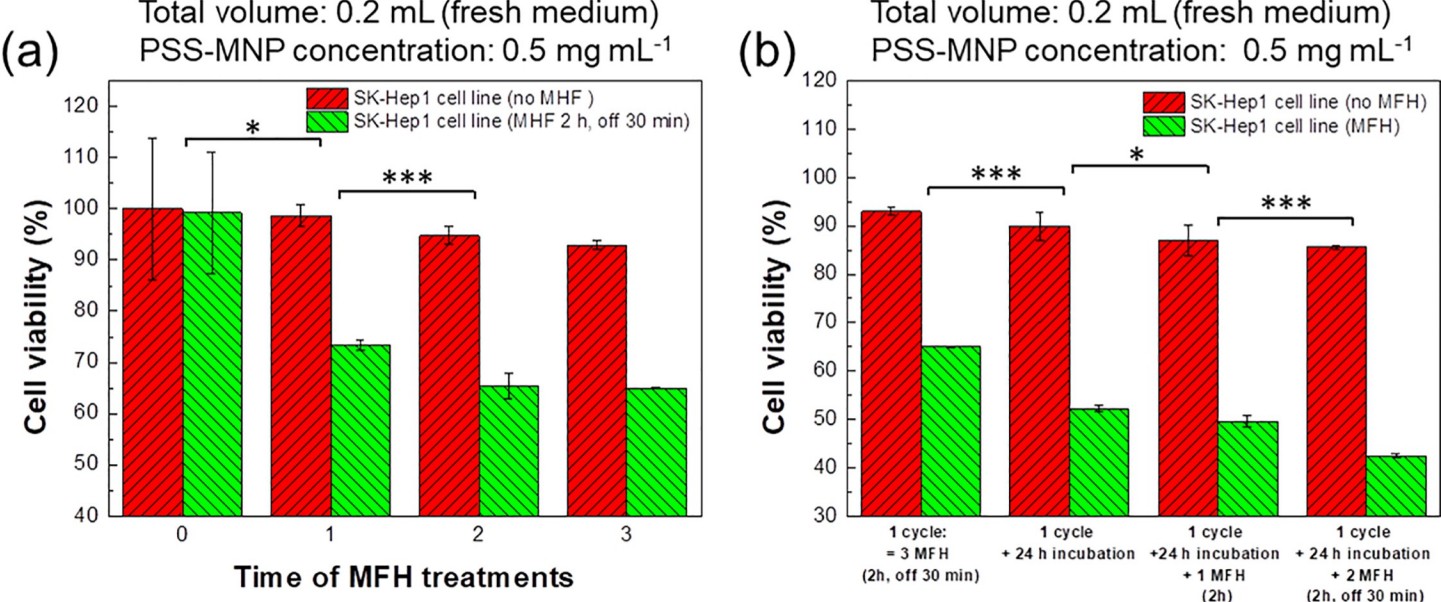

**Fig 6. Intermittent hyperthermia effect on PSS-MNP intracellular hyperthermia therapies with varied operating duration.** The duration was varied from (a) a few operations of MFH in one day (0–3 times) to (b) numerous MFH treatments operated within two days (3 times + incubation 24 h + 0–2 times next day). SK-HEP1 cells were treated with PSS-MNPs (0.5 mg mL$^{-1}$) for cell endocytosis for 24 h, followed by washing with PBS to remove excess MNPs before AMF induction. The fresh culture medium volume was adjusted to 0.2 mL. (a) MNP-treated cells were treated for 2 h with an AMF of magnetic field 157 G and frequency 43.7 kHz, and then rested for 30 min before the next AMF operating cycle. (b) After AMF treatment three times of a first administration, the cells were treated with PSS-MNPs (0.5 mg mL$^{-1}$) as a second administration for 24 h; intermittent hyperthermia steps were further treated and implemented for a few cycles. The degree of significance is given as $^{*}$ $p{<}0.05$; $^{**}$ $p{<}0.01$ and $^{***}$ $p{<}0.001$.

SK-Hep1 cells can still not be ignored and threaten human health; tumor treatments to kill SK-Hep1 cells without side effects must still be developed.

Based on the PSS-MNP characterization in previous work, the PSS-MNPs were biocompatible in human HCC SK-Hep1 and mouse NIH-3T3 cells [17]. In the present study, the PSS-MNPs magnetized with an AMF released thermal energy, which performed lethal thermal effects on the SK-Hep1 cells for *in vitro* examination. In a summary of the above results, the (1) AMF efficiency and (2) volume of the solvent medium are confirmed to affect the MNP-induced lethal thermal effects on the SK-Hep1 cells in the MFH treatments; the optimal hyperthermia parameters were determined under the operating conditions AMF frequency 43.7 kHz and solvent volume 0.2 mL in our self-made AMF machine (Figs 1 and 5). The magnetized PSS-MNPs exhibited significant intracellular hyperthermia effects on SK-Hep1 cells; as the operating period and duration increased, the lethal effects accumulated and enhanced to kill the MNP-treated tumor cells (Fig 6).

In our experimental setup, the components for our self-made hyperthermia machine were purchased commercially and assembled without further remodeling or modification (Fig 1A and S1 Fig). Compared to a commercial hyperthermia operation, the maximum effective AFM generated in our setup achieved 143.5 G under operation at 43.7 kHz; these power and generated magnetic field were less than those in previous studies (magnetic field intensity >200 G, >15 kA m$^{-1}$) [23, 24]. The AMF increased, notably, as the operating frequency decreased in our setup, the reverse of the performance of commercial hyperthermia products: the AMF intensity normally increases as the operating frequency increases [11]. The integration of heat from MNP-induced Néel relaxation under a large external magnetic field and MNPs hence

forced oscillations at a small operating frequency, which are both in areas of extreme opposites, to become a prospective topic of studies of MNP-based tumor therapy.

The extracellular hyperthermia treatments in SK-Hep1 cells (Fig 2D) exhibited a significantly more effective lethal effect than an intracellular treatment (Fig 5A). In extracellular hyperthermia treatment groups (Fig 4), the SK-Hep1 cells exhibited a ductile appearance and the intact DAPI-stained nuclei under microscopic observation before the MFH treatment (Fig 4A). After the MFH treatment for 1 h, the hyperthermia-treated cells in Fig 4B(ii, iii) showed less cell ductility of SK-Hep1 cells, which tended to aggregate to each other, in comparison of Fig 4A. The cell proliferation and growth are considered to have been affected by the AMF-induced MNPs. Some DAPI-stained cells exhibited cell nuclei of broken shapes (Fig 4B(v, vi)) with cell size decreased from 11.36 to 8.99 μm on average (Fig 4C) after the hyperthermia treatments, indicating a possible thermal or mechanical effect induced from PSS-MNPs to the SK-Hep1 cells.

In an evaluation of the heating efficiency of PSS-MNPs, they (concentration 30 mg mL$^{-1}$) with MFH treatment under AMF operation at frequency 43.5 kHz and 143.5 G showed SAR 4.9 W g$^{-1}$ and ILP 0.885 nH m$^2$ kg$^{-1}$ (Fig 1C). The small SAR and ILP values were predictable in the MFH operations, because of AMF operating frequency <100 kHz. Despite an ineffective heating rate, with a greater duration of MFH treatment, a temperature >43 °C was still achieved (within 30 min, Fig 1F), and resulted in the apoptosis and death of SK-Hep1 cells.

For an evaluation of the mechanical effect, as previously stated, the debris of cell nuclei were observed after MFH treatments (Fig 4B(v, vi)). Similar results–the treated cells exhibiting round and shrunken shapes with shattered cell nuclei–have been reported by previous authors concerning tumor therapies; among previous research on the apoptosis induced by MNPs or magnetic microstructures, Kim et al. and Zhang et al. utilized low-frequency AMF [25] and a dynamic magnetic field [26], respectively, to induce a mechanical force or torque inside the cytoplasm through the induction of magnetic components bound to the specific receptors (on the cell membrane) or lysosome (inside the membrane). According to Kim et al., the *N*10 glioma cancer cells treated with gold-coated (thickness 5 nm) iron–nickel permalloy discs (20:80, thickness 60 nm, diameter ~1μm) showed a morphology difference between before and after low-AMF operation (frequency 20 Hz, intensity 90 G, duration 10 min), observed with an atomic force microscope: the cell morphology of treated *N*10 cells altered from a ductile and elongated appearance with clear and distinct nuclei to a round, flat and nucleus-fractionated mochi-like cell, indicating that cell disruption and apoptosis were activated by the magnetized permalloy-disc-induced torque that caused cell death [25]. Also, Zhang et al. prepared lysosomal-associated membrane protein 1 (LAMP1) antibodies conjugated with superparamagnetic iron-oxide nanoparticles and studied how the mechanical force contributed from the MNP rotation affected and disrupted the lysosome intracellularly in both rat insulinoma tumor cells and human pancreatic beta cells. To control the MNPs remotely, a dynamic magnetic field was used to magnetize the MNPs and applied a mechanical force in the intracellular environment; because of the weak dynamic magnetic field used (intensity 300 G, frequency 20 Hz, duration 20 min), the generated heat was ensured to have been contributed from the MNP rotations (frictional force) and Néel relaxation was negligible, so that the efficacy and effects of mechanical force on the tumor killing were precisely studied. Posterior to the operation of the dynamic magnetic field, a shrinkage of lysosome size in human pancreatic beta cells was achieved, indicating that the disruption of lysosome was induced by the MNP rotations in a dynamic magnetic field [26]. According to other studies concerning high-frequency AMF hyperthermia, treated cells had normally decreased cell sizes and altered their shapes to round after MFH; intracellular nuclei and organelles are typically deformed irregularly but maintained an intact appearance after thermal treatment above 43°C, followed by apoptosis. A

decreased cell size was observed after MFH treatments [27, 28]. The above references hence indicate that cell disruption is the main characteristic of cells destroyed by MNP mechanical effects; the hyperthermia effects induced cell deformation irregularly. Both mechanical and thermal effects affect and decrease the cell ductility and size. Our present work showed that most treated SK-Hep1 cells underwent nuclear deformation and others exhibited nuclear disruption, according to DAPI-stained images (Fig 4B), proving that MFH treatments with low-alternating-magnetized PSS-MNPs assisted with our AMF machine exhibited both mechanical and thermal effects contributing to killing the SK-Hep1 cells.

The most challenging issue in future applications is to control the heat surrounding the treated HCC cells or the liver organs: (1) the intracellular thermal effects of PSS-MNPs in a large volume of medium were inadequate to kill SK-Hep1 cells in *in vitro* tests (Fig 5A), *i.e.* there is a hesitation whether the AMF-induced heat generated would be removed by the human blood flow during *in vivo* tests in the future, and (2) the safety of PSS-MNPs in an animal body (*in vivo*), because of a lack of antibody on the MNPs, must still be confirmed. Despite an improved MFH efficiency obtained with a decreased volume of medium and intermittent treatment (Figs 5B and 6), the efficacy on SK-Hep1 cells with PSS-MNPs could be improved, having attained 43% cell viability in a two-day therapy, but the real cases in *in vitro* tests definitely differ from those in *in vivo* treatments. Blood vessels in the tumor region have been proved to cause irregular cooling around the tumor sites [2]; the tumor size significantly affects the heating rate and the SAR values [29], and low-frequency AMF (<100 kHz) could enhance and benefit the metabolism and blood circulation with negligible heat in the absence of MNPs [30]. The decreased heating efficiency of PSS-MNPs in animal bodies could be predicted. Contrary to concerns about cooling effects from blood flow, the heating of tissues at distance 2 cm from the tumor that attained temperature 43 °C in thermotherapy is prohibitive [31]; it is essential that the tumor sites are confirmed with accurate non-invasive techniques before *in vivo* tests, such as MRI assisted with computerized tomography (CT) or ultrasound imaging. It thus becomes a trend that instruments for centralized magnetic-field alignments are equipped or cooperate with imaging instruments as a series of systematic equipment integration, such as magnetic-resonance thermometry (MRT), to implement an efficient MNP accumulation in the tumor sites [3]. Thus, based on the above-mentioned conditions and predictable problems, the cooperation of (1) one-day endocytosis of MNPs for intratumoral treatments and (2) extracellular hyperthermia immediately before an injection into the solid-tissue tumor sites are superior plans for treatments using PSS-MNPs. In case that PSS-MNPs exhibit a non-uniform distribution surrounding not only tumor cells but also in normal tissues, the integration of PSS-MNPs with phospholipid-bilayered carriers conjugated with a specific antibody is one candidate to improve tumor specificity and an effective administered concentration to tumor cells [32].

## Conclusion

We assembled a facile hyperthermia system to conduct low-frequency AMF hyperthermia of SK-Hep1 HCC cells with PSS-MNPs. The AMF system can provide magnetic field 143.5 G with alternating frequency 43.7 kHz, which could be predicted to impose lethal effects on the tumor cells with integration of magnetic hyperthermia and mechanical force. According to the results, the PSS-MNPs have significant thermal therapeutic effects in the extracellular magnetic heat treatment rather than in intracellular magnetotherapy. Decreasing the volume of culture medium improved the lethal effects of intracellular MFH treatment. Moreover, intermittent treatments can further increase the mortality of cancer cells to achieve the same level of effect as extracellular magnetotherapy. Although the results of the *in vitro* MFH tests in SK-Hep1

HCC cells exhibited positive consequences, the primary results in cell-level *in vitro* experiments are only a reference for the ideal state of PSS-MNP thermotherapeutic applications; one must still take into account the size of the tumor and the three-dimensional structure of the state in the animal body. The MFH *in vivo* treatment efficacy depends on the MNP properties (magnetic and biomedical properties), extrinsic parameters (equipment quality and efficiency) and assistance of imaging (determination of tumor and nanoparticle distribution). The improvement and optimization of an AMF machine integrated with diagnostic tools and surgical techniques using PSS-MNPs are hence essential for clinical applications in HCC tumor-cell thermotherapy.

## Supporting information

**S1 Fig. Illustration and photograph of a magnetothermal setup and a test of its thermal heating.** In the thermal control tests, a dish (30 mm) of MNP/medium mixture was placed on the copper-coil-wrapped iron core ($N$ = 100), followed by power-supply operation to generate an AC magnetic field. The AMF frequency was obtained from an oscilloscope. The temperature of the MNP/medium mixture was detected with a thermocouple. To examine the MNP hyperthermia efficiency, we set the water bath to 23 $^o$C to compare with a heated medium temperature.
(TIF)

**S2 Fig. Transmission electron microscope (TEM) image, mono-particle size distribution and thermogravimetric analysis (TGA) of PSS-MNPs.** TEM (Philips field-emission, TEC-NAI 20, electron gun ZrO/W(100) Schottky type, resolution $\leq$ 0.23 nm, Philips, Holland) and thermal analyzer (Mettler-Toledo 2-HT, Switzerland) were used for characterization. (a) A TEM image of PSS coated MNPs is shown; (b) the individual particle sizes were in the range of tens nm; scale bar 100 nm. (c) Thermal analyses of PSS-MNPs was implemented from 23 $^o$C to 800˚C at rate 10 $^o$C min$^{-1}$ in an alumina ($Al_2O_3$) pan under a nitrogen atmosphere. The content of coated polymer, considered as PSS, was 14.1% of the whole PSS-MNPs.
(TIF)

**S3 Fig. *In vitro* examination of human SK-Hep1 HCC and mouse NIH-3T3 fibroblast cell lines to evaluate the cytotoxic effects of PSS-MNPs on cancer and normal cells.** The control groups indicate that the human SK-Hep1 HCC and mouse NIH-3T3 fibroblast cell lines ($5 \times 10^5$ cell mL$^{-1}$) were treated with medium only; AMF groups mean that the cell lines were treated with an AC magnetic field without added MNPs for 40 min; PSS-MNP co-culture groups indicate that the cell lines were co-cultured with PSS-MNPs (0.4 mg mL$^{-1}$) without applied AMF for 40 min; MFH groups indicate that the cells were added with PSS-MNPs (0.4 mg mL$^{-1}$), followed by AMF operation (43.7 kHz, 143.5 G) for 40 min.
(TIF)

**S4 Fig. Analyses of hydrodynamic diameter distribution of PSS-MNPs in varied simulated physiological microenvironment with DLS measurement (Brookhaven 90 plus particle-size analyzer, red diode laser, 40 mW, nominal wavelength 640 nm, Brookhaven instruments Corp., USA).** (a) PSS-MNPs suspended in extracellular (400 µL $H_2O_2$/PBS solution, pH 6.8) and intracellular (10 mM GSH/EDTA solution, pH 5.0) condition. The GSH concentration in hepatocytes is normally 10 mM greater than the concentration in most cells (1–2 mM) [33]. (b) On adding a small amount of negatively charged compounds (25 mM sodium citrate in PBS), the zeta potential values of MNPs were improved from -21.16 mV to -25.98 mV, indicating greater electrostatic interaction of PSS-MNPs, associated with nanoparticle dispersity,

exhibited in PBS; the smaller hydrodynamic diameter distribution was thus attained.
(TIF)

**S1 Table. Summary of PSS-MNP properties from our previous work [17], including saturation magnetization, average core size of individual particles, average hydrodynamic diameter and quantification of cellular uptake mass and ratio in the human SK-Hep1 and mouse NIH-3T3 cells.**
(DOCX)

## Author Contributions

**Conceptualization:** Hao-Ting Huang, Da-Jeng Yao.

**Data curation:** Bo-Wei Chen, Guo-Wei Chiu, Yun-Chi He, Chih-Yu Huang.

**Formal analysis:** Bo-Wei Chen, Yun-Chi He.

**Investigation:** Shian-Ying Sung, Chia-Ling Hsieh, Wei-Chieh Chang, Ming-Shinn Hsu, Zung-Hang Wei.

**Methodology:** Shian-Ying Sung, Zung-Hang Wei.

**Project administration:** Wei-Chieh Chang, Zung-Hang Wei, Da-Jeng Yao.

**Supervision:** Ming-Shinn Hsu.

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
