## [Decision Letter · Decision Letter 0]

29 Sep 2020

PONE-D-20-26378

Polystyrene sulfonic acid coated magnetic nanoparticles based extracellular and intracellular intermittent magnetic fluid hyperthermia treatment on the SK-Hep1 hepatocellular carcinoma cells

PLOS ONE

Dear Dr. Yao,

Thank you for submitting your manuscript to PLOS ONE. After careful consideration, we feel that it has merit but does not fully meet PLOS ONE’s publication criteria as it currently stands. Therefore, we invite you to submit a revised version of the manuscript that addresses the points raised during the review process.

We look forward to receiving your revised manuscript.

Kind regards,

Yi Cao

Academic Editor

PLOS ONE

Journal Requirements:

2. Please amend either the title on the online submission form (via Edit Submission) or the title in the manuscript so that they are identical.

Reviewers' comments:

Reviewer's Responses to Questions

**Comments to the Author**

1. Is the manuscript technically sound, and do the data support the conclusions?

Reviewer #1: Yes

Reviewer #2: Yes

2. Has the statistical analysis been performed appropriately and rigorously? 

Reviewer #1: Yes

Reviewer #2: Yes

3. Have the authors made all data underlying the findings in their manuscript fully available?

Reviewer #1: Yes

Reviewer #2: Yes

4. Is the manuscript presented in an intelligible fashion and written in standard English?

Reviewer #1: Yes

Reviewer #2: Yes

5. Review Comments to the Author

Reviewer #1: The manuscript was mainly focused on polystyrene-sulfonic-acid-coated magnetic nanoparticles and the magnetic fluid hyperthermia treatment on tumor cells. The study was interesting and some meaningful for clinical cancer thermo-treatment application. In addition, the synthesis process, characteristic and evaluation in vitro was abundant and well-founded. Generally the paper is recommended for publication in PLOS ONE after minor revision. There are several questions as follow:

1. All the manuscript should better be checked carefully with some problems with grammar and narration.

2. Compared with other researches, what's the advantage of the author's work? For the copolymer polystyrene sulfonic acid, what role does it play? Are there some other MNPs coated by the same copolymer or different copolymer reported? The author is suggested to enrich the "Introduction" or "Discussion" section, and this may highlight the significance and innovation of this work.

3. How is the stability of the polymer coated magnetic nanoparticles at different simulated physiological microenvironment, such as in bloodsteam (pH 7.4) or extracellular (pH 6.8 with H2O2) and intracellular (pH 5.0 with GSH) condition at tumor site? This was important for the following clinical application.

4. The size distribution of the magnetic nanoparticles was not some well, are there some other methods to improve the size distribution?

5. For organic and inorganic system, the author was suggested to provide the thermogracimetric analysis and elemental analysis data or some other test to prove the successful preparation of the polymer coated magnetic nanoparticles, but not just with the TEM and DLS results.

6. The MFH treatment were just carried out in vitro on cancer cell line level. It maybe of more convincing that the in vivo test and evaluation for cancer treatment are implemented on animal experimental model in the future.

Reviewer #2: In this manuscript, Chen and coworkers examined their polystyrene-sulfonic-acid- coated magnetic nanoparticles (PSS-MNPs) -induced magnetic fluid hyperthermia (MFH) on SK-Hep1 hepatocellular carcinoma (HCC) cells for lethal thermal effects with a self-made AMF system. I suggest acceptance of this manuscript after addressing the following comments.

1. All figures need to be 300dpi. 2. label details in figure S2. 3.what are effects of PSS-MNPs MFH on normal cells？

6. PLOS authors have the option to publish the peer review history of their article (what does this mean?). If published, this will include your full peer review and any attached files.

Reviewer #1: No

Reviewer #2: No

---

## [Author Response · Author response to Decision Letter 0]

3 Dec 2020

Dear editor(s) and reviewers:

Thanks for the valuable advices and suggestions from reviewers. We have added the detailed descriptions, information and the supplemented comparative results of PSS-MNPs including thermogracimetric analysis and in vitro tests (tumor versus normal cells) in the new manuscript. Our revised manuscript based on reviewers’ comments is summarized with a list of the changes in the following pages. Thank you for your time and help in this manuscript.

---

## [Decision Letter · Decision Letter 1]

26 Dec 2020

Extracellular and intracellular intermittent magnetic-fluid hyperthermia treatment of SK-Hep1 hepatocellular carcinoma cells based on magnetic nanoparticles coated with polystyrene sulfonic acid

PONE-D-20-26378R1

Dear Dr. Yao,

We’re pleased to inform you that your manuscript has been judged scientifically suitable for publication and will be formally accepted for publication once it meets all outstanding technical requirements.

Kind regards,

Yi Cao

Academic Editor

PLOS ONE

Additional Editor Comments (optional):

The authors may need to provide better images if needed.

Reviewers' comments:

Reviewer's Responses to Questions

**Comments to the Author**

1. If the authors have adequately addressed your comments raised in a previous round of review and you feel that this manuscript is now acceptable for publication, you may indicate that here to bypass the “Comments to the Author” section, enter your conflict of interest statement in the “Confidential to Editor” section, and submit your "Accept" recommendation.

Reviewer #1: All comments have been addressed

Reviewer #2: (No Response)

2. Is the manuscript technically sound, and do the data support the conclusions?

Reviewer #1: Yes

Reviewer #2: Yes

3. Has the statistical analysis been performed appropriately and rigorously? 

Reviewer #1: Yes

Reviewer #2: Yes

4. Have the authors made all data underlying the findings in their manuscript fully available?

Reviewer #1: Yes

Reviewer #2: Yes

5. Is the manuscript presented in an intelligible fashion and written in standard English?

Reviewer #1: Yes

Reviewer #2: Yes

6. Review Comments to the Author

Reviewer #1: The authors have adequately addressed the reviewer's comments raised in the previous round and the reviewer think that this manuscript is now acceptable for publication without further revision.

Reviewer #2: (No Response)

7. PLOS authors have the option to publish the peer review history of their article (what does this mean?). If published, this will include your full peer review and any attached files.

Reviewer #1: No

Reviewer #2: No

---

## [Editor Report · Acceptance letter]

12 Jan 2021

PONE-D-20-26378R1 

Extracellular and intracellular intermittent magnetic-fluid hyperthermia treatment of SK-Hep1 hepatocellular carcinoma cells based on magnetic nanoparticles coated with polystyrene sulfonic acid 

Dear Dr. Yao:

I'm pleased to inform you that your manuscript has been deemed suitable for publication in PLOS ONE. Congratulations! Your manuscript is now with our production department. 

Kind regards, 

on behalf of

Dr. Yi Cao 

Academic Editor

PLOS ONE